# Neutrophil to lymphocyte ratio as an early indicator for ureteral catheterization in patients with renal colic due to upper urinary tract lithiasis

**Nimer Elsaraya**[1], **Adi Gordon-Irshai**[2], **Dan Schwarzfuchs**[3], **Victor Novack**[2], **Nicola J. Mabjeesh**[1]*, **Endre Z. Neulander**[1]

1 Department of Urology, Soroka University Medical Center, Faculty of Health Sciences, Ben-Gurion University of the Negev, Be'er-Sheva, Israel, 2 Soroka Clinical Research Center, Soroka University Medical Center, Faculty of Health Sciences, Ben-Gurion University of the Negev, Be'er-Sheva, Israel, 3 Emergency Department, Soroka University Medical Center, Faculty of Health Sciences, Ben-Gurion University of the Negev, Be'er-Sheva, Israel

* mabjeesh@bgu.ac.il

**Data Availability Statement:** All relevant data are within the manuscript. However, according to the National laws and regulations the data can't be

## Abstract

### Purpose

To evaluate whether the neutrophil-to-lymphocyte ratio (NLR) can predict the need for ureteral catheterization in patients with renal colic.

### Materials and methods

We retrospectively studied 15,887 patients with renal colic between 2005 and 2019. Patients with prior antibiotics treatment (156), with hematological diseases (15), with negative computerized tomography scan (CTS) for stone disease (473) or with no available laboratory findings (1750) were excluded. A ureteral double J stent (DJS) was inserted in case of ongoing pain, fever, sepsis, single kidney and elevated blood creatinine levels concomitant with hydronephrosis. A cut-off value of 2.1 NLR was determined to stratify and to compare patients using multivariable logistic regression models. A locally weighted scatterplot smoothing (LOWESS) plot was also applied to show the relationship between NLR and predicted probability for DJS insertion.

### Results

Thirteen-thousand and 493 patients with a mean age of 42.7 years (30% females and 70% males) were included in the study. Five-hundred and 57 patients (4.1%) underwent early DJS insertion: 5.3% vs. 1.5% of patients with high vs. low NLR, respectively, ($p<0.001$). High NLR was significantly associated with longer hospitalization time, admission to the intensive care unit and overall mortality within a month from admission ($p<0.05$). LOWESS plot showed that NLR value >2.1 escalates progressively the probability for DJS insertion.

uploaded to the data repository. The data can be shared upon the request addressed to Prof. Eitan Lunenfeld, MD, MHA, Head of IRB, Soroka University Medical Center, Beer-Sheva, Israel. eitan_l@clalit.org.il.

**Funding:** The author(s) received no specific funding for this work.

**Competing interests:** The authors have declared that no competing interests exist.

## Conclusions

A high NLR is associated with the need for early internal DJS insertion due to urolithiasis. The NLR is easily calculated from simple blood tests and based on our results can be used for clinical decision making in patients with renal colic needing renal decompression.

## Introduction

Acute renal colic caused by urolithiasis is a common urological emergency often requiring hospitalization and in some cases, surgical intervention. Emergent surgical decompression of the upper urinary tract in this group of patients is required if there is evidence of sepsis, deteriorating renal function, obstruction of a solitary kidney or ongoing pain despite adequate analgesia [1]. In these cases the recommended method of intervention is either retrograde ureteral double J stent (DJS) insertion or percutaneous nephrostomy with definitive management of the stone after infection and/or obstruction was alleviated.

Despite established clinical and imaging criteria for urgent intervention/ decompression of the upper urinary tract, few studies have looked at factors to identify patients who will require emergency surgery at the time of their initial presentation helping to early predict the need for intervention, before the full-blown clinical picture becomes evident. Serum C-reactive protein (CRP), an acute phase protein, has been used with success to identify those with upper urinary tract infection [2].

A recent study suggested that CRP measured at the time of presentation with acute renal colic could be used to predict who would require urgent surgical intervention [3]. Serum CRP > 28 mg/L at presentation identified those patients who needed emergency intervention with a positive predictive value of 87.2% [3].

Among other inflammatory markers, the neutrophil-to-lymphocyte ratio (NLR), defined as the ratio of absolute counts of neutrophils and lymphocytes, is a simple and effective marker that reflects an imbalance in inflammatory cells [4]. However, to the best of our knowledge, no data have linked NLR to the need for emergent decompressive intervention in cases of ureteral obstruction due to stone disease accompanied by infection/inflammatory process. NLR has been proven a valuable tool for prediction of the outcome of patients in the critical care departments or children with febrile urinary tract infection [5, 6]. There have been several studies showing that NLR is a measure of systemic bacterial inflammation and it has been used as a guide to prognosis in community acquired pneumonia [7–9].

The aim of this study was to investigate whether NLR predicts the need for early ureteral decompression in patients with renal colic due to stones in the upper urinary tract.

## Patients and methods

### Patients

In the study, we included patients 18 years of age or older who were admitted to Soroka University Medical Center (SUMC) with the diagnosis of renal colic due to upper urinary tract lithiasis, between January 2005 and May 2019. SUMC is a 1,100 bed tertiary teaching hospital providing care to a population of 1,140,000 of the Southern district in Israel. SUMC ethics committee approved the study and waived informed consent requirements and all methods were performed in accordance with the relevant guidelines and regulations. Participants who had no evaluable laboratory tests, who used antibiotics 3 weeks prior to admission, who had

hematological diseases such as leukemia, pancytopenia, myelofibrosis and myelodysplastic syndrome and who had a negative non-contrast abdominal CT scan (CTS) for upper urinary tract stones, where excluded from the study.

Indications for emergent decompressive intervention of the upper urinary tract were: ongoing pain, fever/sepsis, stone size and location, rising creatinine/obstruction and single kidney. Our default approach is always to initially insert a DJS in patients with renal colic due to stone obstruction. In few cases where DJS insertion was not successful a percutaneous nephrostomy was then performed. Decompression of upper urinary tract was performed at the first admission in all indicated patients.

NLR was calculated based on the laboratory analysis taken at the time of admission to the emergency department, as absolute neutrophil count divided by the absolute lymphocyte count from the first lab result taken at presentation in the emergency department.

### Statistical analysis

Demographic and clinical characteristics were tabulated. Continuous normal distribution variables including mean and standard deviation, continuous variables that were not normally distributed including median and interquartile range (IQR) and categorical variables were presented as percentage.

The study population was stratified into two groups by NLR cut-off value of 2.1 established in a previous study [10]. Comparison between the groups was performed using t-test for continuous variables with normal distribution, Mann Whitney U Test (Wilcoxon Rank Sum Test) for continuous variables with non-normal distribution and Chi-square test for categorical data.

Logistic regression analysis was used for multivariable modeling. We selected variables with clinical and statistical significance ($p < 0.01$ in univariable analysis) to be included into the model. The discriminatory ability of the models to predict ureteral decompression/stenting was evaluated by c-statistics. We further fitted a smoothed curved line with a locally weighted scatterplot smoothing (LOWESS) plot showing the relationship between NLR and predicted probability of DJS insertion. Statistical significance was defined as p-value $< 0.05$. All analyses were carried out using IBM SPSS Statistics software (Version 25).

### Results

Patient allocations are summarized in a consolidated standards of reporting trials (CON-SORT) diagram (Fig 1). A total of 13,493 eligible patients with a mean age of 42.7 years were included in the study (Table 1). Seventy percent were males and 30% females. Between 2015 and 2019, 828,000 patients were admitted to our emergency department making the incidence of acute renal colic 1.9% of all patients referring to our emergency department with a yearly prevalence of 106 per 100,000 persons of the whole population in our Southern district. The patients were divided into two groups according to normal versus high NLR levels, and their demographic, clinical and laboratory characteristics were compared (Table 1). Four percent of the patients presented with fever but there was no significant difference between the groups. Patients with high NLR levels were relatively older (mean age 44.1 vs. 39.1, $p < 0.001$). Patients with high NLR levels had higher proportion of leukocytosis than those with normal NLR. In the NLR group the percentage of patients with elevated CRP was 3.2 compared to 1.8 in the normal NLR group ($p < 0.001$). Although there was statistically significant difference between the groups in blood creatinine and urea levels at admission this difference was non-clinically significant. There was no difference in patients with clinical signs of shock between the groups.

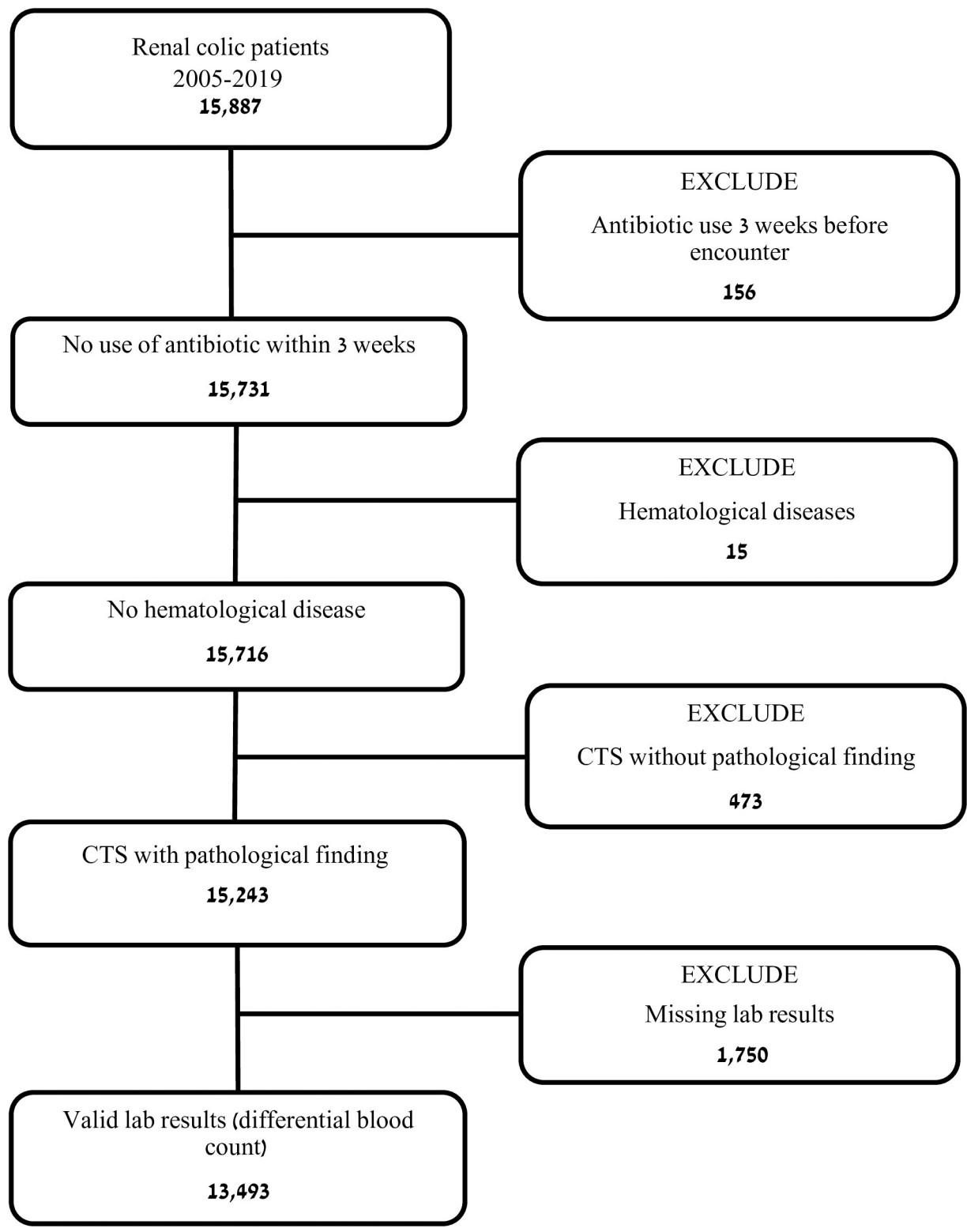

**Fig 1. CONSORT flow diagram of renal colic study patients.**

**Table 1. Demographic and clinical characteristics of renal colic patients.**

| | Total | Normal NLR Group | High NLR Group | P value |
|---|---|---|---|---|
| No. of patients (%) | 13,493 (100) | 4,043 (30) | 9,450 (70) | |
| Gender | | | | |
| Males, No. (%) | 9,446 (70) | 2,788 (69.0) | 6,658 (70.5) | 0.176 |
| Females, No. (%) | 4,046 (30) | 1,255 (31.0) | 2,791 (29.5) | |
| Age (years) | | | | |
| Mean (SD) | 42.66 (15.5) | 39.1 (13.9) | 44.1 (15.8) | < 0.001 |
| Fever > 38˚C | | | | |
| No. (%) | 537 (4) | 168 (4.1) | 369 (3.9) | 0.939 |
| Creatinine (mg/dL) | | | | |
| Mean (SD) | 0.99 (0.55) | 0.89 (0.3) | 1.03 (0.6) | < 0.001 |
| Urea (mg/dL) | | | | |
| Mean (SD) | 34.58 (15.8) | 31.99 (11.8) | 35.67 (17.1) | < 0.001 |
| CRP > 5 (mg/dL) | | | | |
| No. (%) | 379 (2.8) | 74 (1.8) | 305 (3.2) | < 0.001 |
| Leukocytosis >11.0 x10$^3$ | | | | |
| No. (%) | 4,637 (34.4) | 489 (12.1) | 4,148 (43.9) | < 0.001 |
| Shock (high HR, low BP) | | | | |
| No. (%) | 1,295 (9.6) | 405 (10.0) | 891 (9.4) | 0.295 |
| Charlson index | | | | |
| Median (IQR) | 2 (1–4) | 2 (0–3) | 3 (1–4) | < 0.001 |
| Heart failure | | | | |
| No. (%) | 58 (0.4) | 8 (0.2) | 50 (0.5) | 0.007 |
| Diabetes mellitus | | | | |
| No. (%) | 673 (5) | 154 (3.8) | 519 (5.5) | < 0.001 |
| Chronic kidney disease | | | | |
| No. (%) | 143 (1.1) | 24 (0.6) | 119 (1.3) | 0.001 |

NLR, neutrophil to lymphocyte ratio; CRP, c-reactive protein; HR, heart rate; BP, blood pressure; SD, standard deviation; IQR, interquartile range.

Patients' clinical outcome is described in Table 2. All over, 4% of the patient needed upper urinary tract decompression using ureteral DJS insertion. DJS was inserted in 1.5% of the patients with normal NLR compared to 5.3% in patients with high NLR ($p<0.001$). Elevated NLR was associated with higher rates of hospitalization, longer hospitalization period, admission to intensive care units and overall death within a month after admission.

We used a logistic regression model to determine whether NLR can predict the need for ureteral catheterization in renal colic patients. The analysis was performed per each NLR decile with adjustment for age, Charlson's comorbidity index, the presence of renal disease and the presence of inflammation (Table 3). NLR above 2.1 was associated with an increase of the probability for DJS insertion (Table 3) as well as depicted in the LOWESS relationship curve (Fig 2).

## Discussion

In this study we found NLR as a strong predictor for DJS insertion in patients with renal colic due to stone disease in the upper urinary tract. The presence of a ureteral stone is the most common urologic emergency and it can be associated with pain, obstruction of the upper urinary tract and urinary tract infection, as well as fever and urosepsis [1]. Most of ureteral stones

**Table 2. Clinical outcomes.**

|  | Normal NLR Group (n = 4,043) | High NLR Group (n = 9,450) | P value |
|---|---|---|---|
| Ureteral DJS insertion |  |  |  |
| No. (%) | 60 (1.5) | 497 (5.3) | < 0.001 |
| Hospitalization |  |  |  |
| No. (%) | 398 (9.8) | 1,738 (18.4) | < 0.001 |
| Hospitalization Time |  |  |  |
| Median (IQR) | 2 (1–3) | 3 (2–4) | < 0.001 |
| Hospitalization > a week |  |  |  |
| No. (%) | 15 (0.4) | 128 (1.4) | 0.002 |
| Admission to ICU |  |  |  |
| No. (%) | 0 (0.0) | 12 (0.1) | 0.023 |
| Death within a Month* |  |  |  |
| No. (%) | 1 (0.0) | 14 (0.1) | 0.049 |

NLR, neutrophil to lymphocyte ratio; IQR, interquartile range; DJS, double J stent; ICU, intensive care unit

*, relates to overall mortality.

do not require early decompressive intervention in consistent with our study, out of 13,493 patients only 4% required emergent decompression.

Stone size and location are generally considered the most important factors associated with spontaneous ureter stone passage [1]. However, especially in the last decade acute inflammatory markers were evaluated as predictors of spontaneous stone passage [4]. According to the current literature, serum CRP concentration, pyuria, hydronephrosis, and helical CTS findings of perinephric fat stranding and the tissue-rim sign related to inflammatory changes are negative predictors for spontaneous stone passage [4].

The NLR is a parameter that can be used to evaluate the inflammatory status of a patient. NLR could also be utilized for patients with urinary stones as an objective proxy for

**Table 3. Logistic regression model for prediction of ureteral catheterization by NLR (deciles) adjusted for age, Charlson's comorbidity index, presence of renal disease and presence of inflammation.**

|  | NLR limits | OR (95% CI) | P value |
|---|---|---|---|
| 1st NLR decile | < 1.32 |  |  |
| 2nd NLR decile | 1.32–1.73 | 0.88 (0.23–3.39) | 0.862 |
| 3rd NLR decile | 1.73–2.10 | 0.917 (0.24–3.50) | 0.899 |
| 4th NLR decile | 2.10–2.57 | 1.81 (0.58–5.68) | 0.306 |
| 5th NLR decile | 2.57–3.15 | 1.31 (0.39–4.38) | 0.659 |
| 6th NLR decile | 3.15–3.90 | 2.63 (0.86–8.04) | 0.090 |
| 7th NLR decile | 3.94–4.93 | 3.65 (1.22–10.92) | 0.020 |
| 8th NLR decile | 4.93–6.43 | 2.72 (0.88–8.45)) | 0.083 |
| 9th NLR decile | 6.43–9.21 | 5.40 (1.81–16.02) | 0.002 |
| 10th NLR decile | > 9.21 | 6.81 (2.28–20.36) | 0.001 |
| Age, per year |  | 1.02 (1.01–1.04) | < 0.001 |
| Charlson index, per 1 point |  | 1.05 (0.99–1.12) | 0.083 |
| Renal disease* |  | 2.11 (1.30–3.42) | 0.003 |
| Inflammation** |  | 1.12 (0.78–1.61) | 0.544 |

*Renal disease: medical history of chronic kidney disease or renal failure

**Inflammation: CRP > 5 mg/dL or leukocytosis > 11,000.

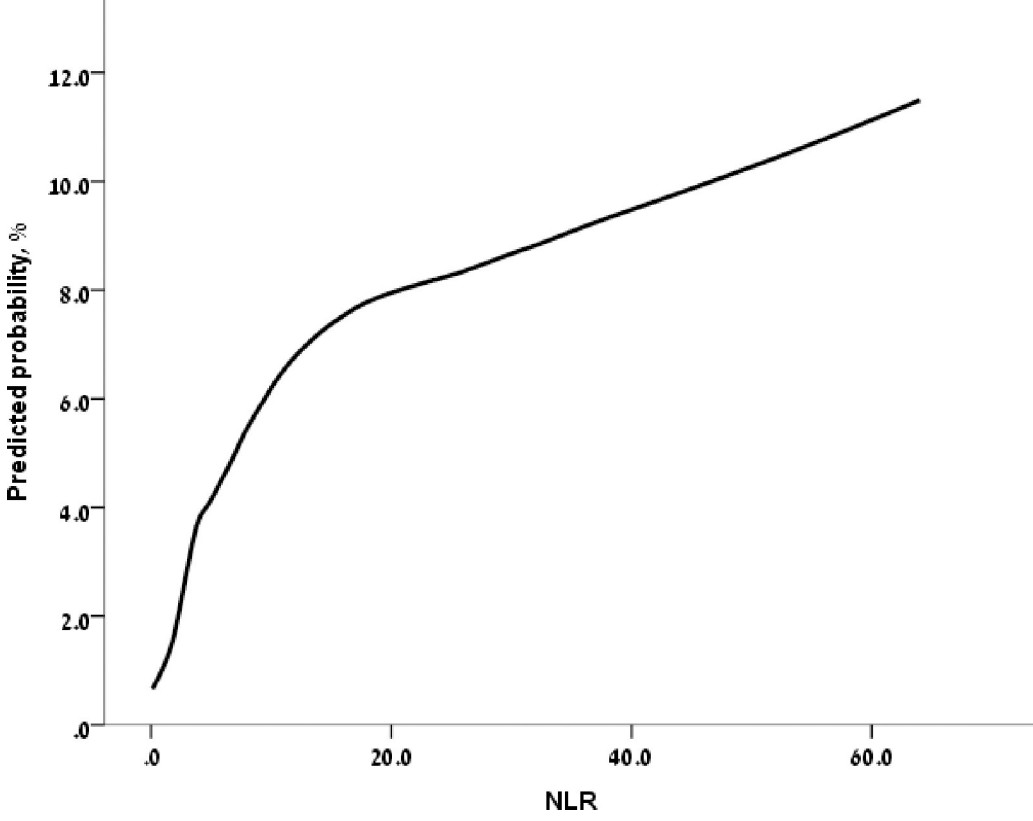

**Fig 2. LOWESS relationship between NLR and predicted probability of ureteral catheterization.**

spontaneous ureter stone passage since inflammatory status has been proven as an important factor in spontaneous stone passage [11, 12]. Forget et al. reported that normal NLR values for non-geriatric adults in good health were between 0.78 and 3.53. In a representative sample of 9,427 subjects in the United States, the average NLR was 2.15 in the general population [10]. Kwang et al. showed that the median NLR of 113 patients in their study was 2.18, and the median NLR in patients who experienced spontaneous ureter stone passage was 2.04 [11]. Patients who did not experience stone passage had a higher NLR (3.67) than those who passed ureteral stones, which supports the notion that inflammation plays an important role in the pathophysiology of spontaneous ureter stone passage [11]. They concluded that size, location of ureter stones and low NLR ($< 2.3$) were independent positive predictors of spontaneous stone expulsion in patients with ureter stones <1.0 cm in size. Therefore, early intervention, rather than expectant management, may be considered for patients presenting with high NLR at initial stone episode as supported by the data presented in this study [11].

Beyond the expression of local upper urinary tract inflammation, NLR has been shown to be related to systemic inflammatory status too. Neutrophils and lymphocytes are the two major inflammatory cell types in the body. NLR is used in a wide range of applications for the diagnosis, treatment, and prognostic evaluation of inflammation-related diseases, such as malignant tumors, cardiovascular diseases, renal diseases, inflammatory bowel diseases and bacteremia. The NLR was also recently described as a marker for the diagnosis of bacterial infections in young infants with febrile urinary tract infection [6, 13]. Gurol et al. showed that according to the level of NLR, predictions can me made regarding local infection, systemic infection and severe sepsis using NLR cutoff value of 5 [14].

In our study, we calculated retrospectively the NLR of 13,493 patients and found significant correlations between NLR and the need for ureteral decompression performed on clinical laboratory and imaging indication. This correlation between NLR and ureteral decompression also remained significant on multivariate logistic regression with adjustment for clinical and laboratory parameters. When considering NLR as a continuous variable we found significant direct correlation between the increasing deciles of NLR value and the increasing probability for ureteral decompression.

Limitations of the study include the retrospective nature of the study and that is not a planned randomized prospective study. However, the number of the patients included is large and spanning over more than a decade and as such, this fact and the sound statistical methods used, may compensate for the inherent flaws of a retrospective study with a smaller sample size. In addition, we did not take into consideration the size and location of stones in the upper urinary tract in our analysis leaving only the reactive severe findings on CTS in the calculations.

## Conclusion

Based on this study we found NLR a reliable marker, independent predictor or indicator for the early decompression of the upper urinary tract in patient presenting to the emergency department with renal colic due to urolithiasis. We believe NLR in addition to clinical, laboratory and imaging studies can play an important role in the decision for early decompression of the upper urinary tract.

## Author Contributions

**Conceptualization:** Nimer Elsaraya, Dan Schwarzfuchs, Victor Novack, Nicola J. Mabjeesh, Endre Z. Neulander.

**Data curation:** Adi Gordon-Irshai, Victor Novack.

**Formal analysis:** Adi Gordon-Irshai, Victor Novack.

**Methodology:** Victor Novack.

**Resources:** Dan Schwarzfuchs.

**Software:** Adi Gordon-Irshai, Victor Novack.

**Supervision:** Victor Novack, Nicola J. Mabjeesh.

**Validation:** Nimer Elsaraya, Dan Schwarzfuchs, Victor Novack, Nicola J. Mabjeesh, Endre Z. Neulander.

**Writing – original draft:** Endre Z. Neulander.

**Writing – review & editing:** Nicola J. Mabjeesh, Endre Z. Neulander.

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
