## [Decision Letter · Decision Letter 0]

29 Oct 2021

PONE-D-21-23297

Neutrophil to lymphocyte ratio as an early indicator for ureteral catheterization in patients with renal colic due to upper urinary tract lithiasis

PLOS ONE

Dear Dr. Mabjeesh,

Thank you for submitting your manuscript to PLOS ONE. After careful consideration, we feel that it has merit but does not fully meet PLOS ONE’s publication criteria as it currently stands. Therefore, we invite you to submit a revised version of the manuscript that addresses the points raised during the review process.

We look forward to receiving your revised manuscript.

Kind regards,

Tzevat Tefik, MD

Academic Editor

PLOS ONE

“ No.The author(s) received no specific funding for this work.”

Reviewers' comments:

Reviewer's Responses to Questions

**Comments to the Author**

1. Is the manuscript technically sound, and do the data support the conclusions?

Reviewer #1: Partly

Reviewer #2: Yes

2. Has the statistical analysis been performed appropriately and rigorously? 

Reviewer #1: Yes

Reviewer #2: Yes

3. Have the authors made all data underlying the findings in their manuscript fully available?

Reviewer #1: Yes

Reviewer #2: Yes

4. Is the manuscript presented in an intelligible fashion and written in standard English?

Reviewer #1: Yes

Reviewer #2: Yes

5. Review Comments to the Author

Reviewer #1: The study seems interesting and up-to-date. If the authors can clarify several issues as outlined below, the article can be more impressive.

- In the article, the NLR cut-off value was determined as 3.5 with reference to a previously published article. In the cited study, the NLR mean value was found to be 1.65 and the upper limit value was 3.53. Also, in their study authors stated that "NLR above 2.1 was associated with an increase of the probability for DJS insertion". Would the results have been different if the NLR cut-off value had been taken as 2.1?

- In the study, it was emphasized that NLR was a predictor for upper urinary tract decompression. In the method, the indications for emergency decompression of the upper urinary system were described. How many patients underwent upper urinary system decompression at the first admission and how many in the following period?

- In Table 2, "Death within a Month" rate in the high NLR group is remarkable. Did these deaths occur due to complications related to upper urinary tract stones or due to other reasons?

Reviewer #2: I would like to congratulate the study about a current issue.The importance of using neutrophil to lymphocyte ratio which is a novel diagnostic biomarker for disease such as kidney stone is increasing recently. I hope that such studies will contribute to the literature. However, I think this study which had a large number of included patients and including more than ten years can be published afer correcting some minor points.

Minor points

-The linguistics of the article should be revised by a native English speaker. Also the article should be checked to correct the spelling errors.

-The incidence and prevalence of Incidence of acute renal colic/nephrolithiasis presenting to the emergency department should also be mentioned in the manuscript.

-Hematological disease or leukemia should be added to the exclusion criteria.

-Also specific contributions by the authors individually part should be added to the manuscript.

- According to the my oppinion, in the new trials, it may be possible to produce more valuable studies by adding information about stone characterization such as stone localization, number, Hounsfield Unit and type of kidney stone.

6. PLOS authors have the option to publish the peer review history of their article (what does this mean?). If published, this will include your full peer review and any attached files.

Reviewer #1: No

Reviewer #2: **Yes: **I would like to congratulate the study about a current issue.The importance of using neutrophil to lymphocyte ratio which is a novel diagnostic biomarker for disease such as kidney stone is increasing recently. I hope that such studies will contribute to the literature. However, I think this study which had a large number of included patients and including more than ten years can be published afer correcting some minor points.

Minor points

-The linguistics of the article should be revised by a native English speaker. Also the article should be checked to correct the spelling errors.

-The incidence and prevalence of Incidence of acute renal colic/nephrolithiasis presenting to the emergency department should also be mentioned in the manuscript.

-Hematological disease or leukemia should be added to the exclusion criteria.

-Also specific contributions by the authors individually part should be added to the manuscript.

- According to the my oppinion, in the new trials, it may be possible to produce more valuable studies by adding information about stone characterization such as stone localization, number, Hounsfield Unit and type of kidney stone.

Dr. Yasin Yitgin

Adress: Istinye Universtiy, Faculty of Medicine, Depatment of Urology, Istanbul, Turkey

e-mail: yasinyitgin@hotmail.com

---

## [Author Response · Author response to Decision Letter 0]

22 Jan 2022

PONE-D-21-23297

Neutrophil to lymphocyte ratio as an early indicator for ureteral catheterization in patients with renal colic due to upper urinary tract lithiasis

Reviewer #1: The study seems interesting and up-to-date. If the authors can clarify several issues as outlined below, the article can be more impressive.

Comment #1

- In the article, the NLR cut-off value was determined as 3.5 with reference to a previously published article. In the cited study, the NLR mean value was found to be 1.65 and the upper limit value was 3.53. Also, in their study authors stated that "NLR above 2.1 was associated with an increase of the probability for DJS insertion". Would the results have been different if the NLR cut-off value had been taken as 2.1?

Response

We agreed with the reviewer and re-analyzed our data taking NLR cut-off value of 2.1 and excluding patients with hematological diseases according to Reviewer’s #2 request. The new results did not change the major conclusions. The new data were changed accordingly throughout the whole manuscript including Tables and Figures. 

Comment #2

- In the study, it was emphasized that NLR was a predictor for upper urinary tract decompression. In the method, the indications for emergency decompression of the upper urinary system were described. How many patients underwent upper urinary system decompression at the first admission and how many in the following period?

Response

Decompression of upper urinary tract was performed at the first admission in all indicated patients. This was clarified under the section of “Patients and Methods” 

Comment #3

- In Table 2, "Death within a Month" rate in the high NLR group is remarkable. Did these deaths occur due to complications related to upper urinary tract stones or due to other reasons?

Response

The rate of death mentioned in in the results and Table 2 relates to overall mortality.  

Reviewer #2: I would like to congratulate the study about a current issue. The importance of using neutrophil to lymphocyte ratio which is a novel diagnostic biomarker for disease such as kidney stone is increasing recently. I hope that such studies will contribute to the literature. However, I think this study which had a large number of included patients and including more than ten years can be published after correcting some minor points.

Comment #1

-The linguistics of the article should be revised by a native English speaker. Also the article should be checked to correct the spelling errors.

Response

The manuscript was re-edited by a native English speaker and rechecked for typos. 

Comment #2

-The incidence and prevalence of Incidence of acute renal colic/nephrolithiasis presenting to the emergency department should also be mentioned in the manuscript.

Response

Between 2015 and 2019, a total of 828,000 patients were admitted to our emergency department making the incidence of acute renal colic 1.9% of all patients referring to our emergency department with a prevalence of 106 per 100,000 persons of the whole population in our Southern district. This info was added to Results accordingly.

Comment #3

-Hematological disease or leukemia should be added to the exclusion criteria.

-Also specific contributions by the authors individually part should be added to the manuscript.

Response

Following the reviewer’s request, we excluded patients with hematological diseased in our re-analysis. The new data were changed accordingly throughout the whole manuscript including Tables and Figures.

The specific contribution of all authors was added to manuscript. 

Comment #4

- According to the my opinion, in the new trials, it may be possible to produce more valuable studies by adding information about stone characterization such as stone localization, number, Hounsfield Unit and type of kidney stone.

Response

We fully agree with reviewer and future studies will include the aforementioned data. This was emphasized in the limitations of the study in the original manuscript.

---

## [Decision Letter · Decision Letter 1]

6 May 2022

PONE-D-21-23297R1Neutrophil to lymphocyte ratio as an early indicator for ureteral catheterization in patients with renal colic due to upper urinary tract lithiasisPLOS ONE

Dear Dr. Mabjeesh,

Thank you for submitting your manuscript to PLOS ONE. After careful consideration, we feel that it has merit but does not fully meet PLOS ONE’s publication criteria as it currently stands. Therefore, we invite you to submit a revised version of the manuscript that addresses the points raised during the review process.

Please pay attention to reviewer 1's comments and ensure all the possible errors are corrected at this time as PLSO ONE does not provide copyediting or proofs of accepted manuscripts. We therefore recommend that you carefully review your manuscript. In addition, PLOS journals require authors to make all data underlying the findings described in their manuscript fully available without restriction at the time of publication (https://journals.plos.org/plosone/s/data-availability>). This policy is aimed to ensure that other researchers can reproduce the analysis. You have stated that "All relevant data are within the manuscript and its Supporting Information files.", but have not provided any supporting information for your data. In light of this, before we can proceed with your submission, please deposit your underlying data for all the tables and figures to a public data repository or include it in the Supporting Information files and update your data availability statement accordingly. If you cannot share your data publicly, for instance, due to privacy or other concerns, please explain why, and include contact information for data requests.

We look forward to receiving your revised manuscript.

Kind regards,

Jianhong Zhou

Staff Editor

PLOS ONE

Journal Requirements:

Reviewers' comments:

Reviewer's Responses to Questions

**Comments to the Author**

1. If the authors have adequately addressed your comments raised in a previous round of review and you feel that this manuscript is now acceptable for publication, you may indicate that here to bypass the “Comments to the Author” section, enter your conflict of interest statement in the “Confidential to Editor” section, and submit your "Accept" recommendation.

Reviewer #1: All comments have been addressed

Reviewer #2: (No Response)

2. Is the manuscript technically sound, and do the data support the conclusions?

Reviewer #1: No

Reviewer #2: Yes

3. Has the statistical analysis been performed appropriately and rigorously? 

Reviewer #1: Yes

Reviewer #2: Yes

4. Have the authors made all data underlying the findings in their manuscript fully available?

Reviewer #1: (No Response)

Reviewer #2: Yes

5. Is the manuscript presented in an intelligible fashion and written in standard English?

Reviewer #1: Yes

Reviewer #2: Yes

6. Review Comments to the Author

Reviewer #1: I thank the authors for preparing this original article. In general, the authors took into account and made the necessary revisions in the article. Only one thing catches my attention in the manuscript. There is probably a numerical mistake in the sentence "Fifty-seven patients (4.1%) underwent early DJS insertion" in the Results of the Abstract section. After the correction of this mistake, it would be more appropriate to accept the article. After the correction of this mistake, the manuscript could be acceptable for publication.

Reviewer #2: I would like to thank the Editor for giving me an opportunity to review the revised paper.

Firstly would like to thank the authors for taking in the suggestions and feedback. It seems the authors have done a lot of work to improve it. Again well done on taking in the feedback and suggestions and working on the manuscript.

7. PLOS authors have the option to publish the peer review history of their article (what does this mean?). If published, this will include your full peer review and any attached files.

Reviewer #1: **Yes: **Samed Verep

Reviewer #2: No

---

## [Author Response · Author response to Decision Letter 1]

28 May 2022

PONE-D-21-23297R1

Neutrophil to lymphocyte ratio as an early indicator for ureteral catheterization in patients with renal colic due to upper urinary tract lithiasis

Reviewer #1: 

Comment #1

Only one thing catches my attention in the manuscript. There is probably a numerical mistake in the sentence "Fifty-seven patients (4.1%) underwent early DJS insertion" in the Results of the Abstract section.

Response

We thank the reviewer for drawing our attention for the numerical mistake appearing in the abstract: writing 57 instead of 557. The mistake was corrected accordingly.

---

## [Editor Report · Decision Letter 2]

16 Jun 2022

Neutrophil to lymphocyte ratio as an early indicator for ureteral catheterization in patients with renal colic due to upper urinary tract lithiasis

PONE-D-21-23297R2

Dear Dr. Mabjeesh,

We’re pleased to inform you that your manuscript has been judged scientifically suitable for publication and will be formally accepted for publication once it meets all outstanding technical requirements.

Kind regards,

Jianhong Zhou

Staff Editor

PLOS ONE
---

## [Editor Report · Acceptance letter]

21 Jun 2022

PONE-D-21-23297R2 

Neutrophil to lymphocyte ratio as an early indicator for ureteral catheterization in patients with renal colic due to upper urinary tract lithiasis 

Dear Dr. Mabjeesh:

I'm pleased to inform you that your manuscript has been deemed suitable for publication in PLOS ONE. Congratulations! Your manuscript is now with our production department. 

Kind regards, 

on behalf of

Jianhong Zhou 

Staff Editor

PLOS ONE